# Changes of Spasticity across Time in Prolonged Disorders of Consciousness: A Retrospective Study

**DOI:** 10.3390/brainsci12020295

**Published:** 2022-02-21

**Authors:** Benjamin Winters, Bruce Kuluris, Rita Pathmanaban, Hannelise Vanderwalt, Aurore Thibaut, Caroline Schnakers

**Affiliations:** 1Department of Psychology, University of California Los Angeles, Los Angeles, CA 90095, USA; bwinters@g.ucla.edu; 2Neurorestorative, CA, USA; bruce.kuluris@sevitahealth.com (B.K.); ritapath@gmail.com (R.P.); hanna.vanderwalt@sevitahealth.com (H.V.); 3Coma Science Group, GIGA Consciousness & Centre du Cerveau, University and University Hospital of Liège, 4000 Liège, Belgium; athibaut@uliege.be; 4Department of Neurosurgery, University of California Los Angeles, Los Angeles, CA 90095, USA; 5Research Institute, Casa Colina Hospital and Centers for Healthcare, Pomona, CA 91767, USA

**Keywords:** spasticity, severe brain injury, disorders of consciousness, minimally conscious state, vegetative state, unresponsive wakefulness syndrome

## Abstract

Objectives: In this retrospective study, we investigated how spasticity developed in patients diagnosed with a prolonged DOC over an almost two-year observation period (21 months), and how it related to the patients’ age, gender, time since injury, etiology, level of consciousness, and anti-spastic medications. Methods: In total, 19 patients with a severe brain injury and prolonged DOC admitted to a long-term care facility were included in this study (14 male, age: 45.8 ± 15.3 years, 10 traumatic brain injury, 1.01 ± 0.99 years after brain injury, 11 minimally conscious state vs. 8 vegetative state). Each patient was assessed at admission and then quarterly, totaling eight assessments over 21 months. Spasticity was measured with the Modified Ashworth Scale (MAS) for both upper and lower limbs. The Western Neuro Sensory Stimulation Profile (WNSSP) was administered to assess the level of consciousness. Any other medical and demographic information of interest was obtained through medical records. Linear mixed models were used to assess each variable’s impact on the change of spasticity over time. Results: Significant differences were observed in the evolution of spasticity in patients based on their etiology for the upper limbs [F (7, 107.29) = 2.226; *p* = 0.038], and on their level of consciousness for the lower limbs [F (7, 107.07) = 3.196; *p* = 0.004]. Conclusion: Our preliminary results suggest that spasticity evolves differently according to the type of brain lesion and the level of consciousness. Spasticity in DOCs might therefore be mediated by different mechanisms and might have to be treated differently among patients. Future longitudinal studies should be performed prospectively in a bigger cohort and with data collection beginning earlier after brain injury to confirm our results and better understand the evolution of spasticity in this population.

## 1. Introduction

Spasticity is best defined as a motor disorder characterized by a velocity-dependent increase in tonic stretch reflexes with exaggerated tendon jerks, resulting from a hyperexcitability of the stretch reflex which results from abnormal intra-spinal processing of primary afferent inputs [1]. Spasticity frequently develops in patients with severe brain injury who are diagnosed with a prolonged disorder of consciousness (DOC). More exactly, the prevalence of spasticity in DOC patients ranged from 59% to 89% in a recent review of 18 published articles [2]. This prevalence is especially concerning since spasticity impacts the patients’ rehabilitation process and quality of life. DOC patients are often bedridden and lack voluntary movements, which, in turn, exacerbates the spastic symptoms [3]. Indeed, functional autonomy relies on the optimal recovery of motor functions after severe brain injury. Previous research has shown that DOC patients with a higher number of medical complications (including spasticity) during inpatient rehabilitation were more likely to exhibit lower functional levels one-year post-injury [4]. The presence of spasticity also complicates behavioral assessments, and can lead to misdiagnosis or underestimation of cognitive functioning [5,6]. Finally, the level of spasticity has been correlated to the level of pain assessed in DOC patients, suggesting that managing spasticity should be a priority for clinicians facing this population [7].

The three main options for treatment are currently physical therapy, antispastic medication, and splints. The primary antispastic medication used in DOC patients is baclofen. Baclofen is a muscle relaxant that can be administered orally or intrathecally, and antagonizes GABA_B_ receptors [8]. Other drugs such as diazepam, clonazepam, gabapentin, or tizanidine, can be used. Botulinum toxin can also be administered to reduce the spasticity of a specific muscle. Despite its broad usage in this population, a review of trials showed that there is limited evidence of effective oral baclofen treatment of spasticity in individuals with a brain injury [9]. Previous research has found that thirty minutes of soft splint application reduces spasticity and improves hand opening of patients with chronic DOCs, and that soft splinting is well tolerated and does not require supervision [10]. However, there has yet to be a published study about the efficacy of using such a treatment over an extended period of time. 

Therefore, despite the high prevalence and impact of this impairment in this population, current treatment options are very limited and inefficacious at treating spasticity in DOC patients. One explanation could be that little is known on how spasticity develops over time and what factors influence its progression [11]. A better understanding of the mechanisms of spasticity would most likely help clinicians in finding treatments that are more efficient to manage spasticity. Therefore, the aim of this retrospective study is to investigate how spasticity develops in patients diagnosed with a prolonged DOC over an almost two-year observation period (21 months), and what medical and demographic factors might affect its development. 

## 2. Methods

### 2.1. Study Population

Retrospective data from March 2010 to July 2016 were collected. The inclusion criteria were as follows: (1) diagnosis of a DOC such as a vegetative state or a minimally conscious state based on the Western Neuro Sensory Stimulation Profile [12], (2) a prolonged DOC (more than 28 days after injury) [13], (3) age 18 years or over, and (4) no documented neurological disorders prior the severe acquired brain injury. All patients were provided nutrition by feeding tube. All patients were admitted to a long-term care facility (Neurorestorative, California) from home, nursing facilities, or other hospitals. 

### 2.2. Study Design

This retrospective longitudinal study was centered on data obtained during physical examinations of patients with DOCs. These examinations were performed quarterly for almost 2 years (i.e., 21 months; for a total of 8 time-points). The spasticity of the upper limbs and lower limbs was assessed by a trained physical therapist using the Modified Ashworth Scale. The Western Neuro Sensory Stimulation Profile was administered to determine the level of consciousness by trained medical staff. For these two outcome measures, data were extracted from scoring sheets in the patients’ medical records by B.K., R.P., H.V., and C.S. The medical and demographic variables of interest were also extracted from patients’ medical records by B.K., R.P., H.V., and C.S.

### 2.3. Outcome Measures

#### 2.3.1. Spasticity

Muscle tone was assessed at the elbow, wrist, and fingers bilaterally for the upper limbs and at the hip, knees, and ankles bilaterally for the lower limbs. The tone assessment followed the procedure of the Modified Ashworth Scale (MAS), a 6-level ordinal scale with documented reliability [14]. Higher MAS scores indicate an increasing severity of spasticity. The assessment of spasticity followed the guidelines of the scale and included passive flexion and the extension of the joints of the upper and lower limbs. The median MAS score of the assessable joints for each limb location from the 8 time-points was used for analyses. 

#### 2.3.2. Level of Consciousness

The level of consciousness was estimated for each patient using the Western Neuro Sensory Stimulation Profile (WNSSP). The WNSSP is used to measure DOCs in patients with a severe acquired brain injury who are slow to recover, and demonstrates very high internal consistency, reliability, and concurrent validity [12]. This scale has been recommended (with moderate reservation) by the American Congress of Rehabilitation Medicine (ACRM) to assess the level of consciousness in DOC patients [15]. The WNSSP has 6 subscales assessing arousal, responses to olfactory, tactile, auditory, and visual stimuli, as well as communication with a total score ranging from 0 to 113. All patients were categorized as being in either a minimally conscious state (MCS) [13] or a vegetative state (VS) [16], also known as unresponsive wakefulness syndrome (UWS) [17], based on their behavioral profile on the scale at the first time-point.

#### 2.3.3. Medical and Demographic Data

The etiology was determined based on medical records. Patients were considered to have a DOC of traumatic (T) etiology if they had a traumatic brain injury, and of non-traumatic (NT) etiology if they had a (hemorrhagic and ischemic) stroke or anoxia. The time since injury (TSI) was calculated for each patient as the number of days between the initial date of injury and the first time-point. The use of antispastic medication (e.g., baclofen) was tracked through each recording session for all patients. Since there was attrition, a ratio was created to address the presence of antispastic medications during the observation period by dividing the number of trials where medication was present by the total number of trials. This resulted in a score of one if the patient was taking an antispastic medication at every session and zero if no medication was used. Finally, patients’ gender (male/female) and age at the first time-point were also recorded.

### 2.4. Statistical Analysis

To assess the changes in spasticity over time, a linear mixed model (LMM) was used to assess the significance of grouping factors on the MAS score. The longitudinal LMM analysis had the advantage of handling covariance among repeated measures. Comparison groups were created between the MAS score and each of the factors previously described, that is: level of consciousness, etiology, TSI, medication, gender, and age. All results were considered significant at *p* < 0.05.

## 3. Result

### 3.1. Participants

This study included 19 patients with a severe brain injury admitted to a long-term care facility (14 male, age: 45.8 ± 15.3 years, 10 traumatic brain injury, 11 within a year of brain injury, 11 minimally conscious state) (Table 1). Ten patients were receiving antispastic medication at the first time-point. The only antispastic medication prescribed to patients was baclofen.

For each patient, one assessment took place on admission before assessments were performed quarterly, totaling eight time-points in almost two years (21 months). The attrition from the eight measurements was as follows: measurements 1–5 (*n* = 19), measurement 6 (*n* = 17), measurement 7 (*n* = 15), and measurement 8 (*n* = 13). Attrition was due to the transfer of patients to another long-term care facility or discharge home. This resulted in a total of 140 MAS measurements across patients.

### 3.2. Upper Limb Spasticity and Brain Injury Etiology 

A significant difference was observed in the evolution of spasticity in patients based on their etiology for the upper limbs (F (7, 107.29) = 2.226; *p* = 0.038). A significant interaction was found between etiology and the repeated MAS measurements, where the traumatic etiology had less upper limb spasticity than the non-traumatic etiology at the beginning of the observation period. No other significant results were found for upper limb spasticity (see Table 2 and Figure 1).

To assess the differences in patients’ characteristics between groups (T/NT), a *t*-test was used for continuous variables (i.e., age and time since injury), and a Chi-square test for dichotomic variables (i.e., gender and level of consciousness). We did not find any differences in terms of age (*t* = −0.727; *p* = 0.477), time since injury (*t* = 1.636; *p* = 0.120), medication (*t* = −0.167; *p* = 0.870), gender (χ^2^ = 2.898; *p* = 0.089), or level of consciousness (χ^2^ = 2.773; *p* = 0.096) between groups (T/NT) for upper limb spasticity at the first time-point.

### 3.3. Lower Limb Spasticity and Level of Consciousness

A significant difference was observed in the evolution of spasticity in patients based on their level of consciousness for the lower limbs (F (7, 107.07) = 3.196; *p* = 0.004). A significant interaction was found between DOCs and the repeated MAS measurements of the lower limb where VS patients developed more spasticity in the second year of observation. No other significant results were found for lower limb spasticity (see Table 3 and Figure 2). 

To assess differences in patients’ characteristics between groups (VS/MCS), a *t*-test was utilized for continuous variables (i.e., age and time since injury) and a Chi-square test for dichotomic variable (i.e., gender and etiology). We did not find any difference in terms of age (*t* = −0.067; *p* = 0.948), time since injury (*t* = 0.401; *p* = 0.693), medication (*t* = −0.936; *p* = 0.362), gender (χ^2^ = 0.012; *p* = 0.912), or etiology (χ^2^ = 2.773; *p* = 0.096) between patients in the MCS and the VS groups for lower limb spasticity at the first time-point.

## 4. Discussion

In this retrospective study, we investigated how spasticity developed in patients diagnosed with a prolonged DOC over an almost two-year observation period (21 months), and how it related to the patients’ age, gender, time since injury, etiology, level of consciousness, and medication. We found two potentially important factors that mediate the development of spasticity in this dataset: etiology of the brain injury for the upper limbs, and the level of consciousness for the lower limbs.

For the upper limbs, patients with traumatic etiologies seemed to build spasticity more gradually than patients with non-traumatic etiologies. As per our statistical analyses, this difference was not related to other factors such as age, time since injury, the presence of spastic medication, or the level of consciousness at the first time-point. Spasticity is related to lesions of the upper motor neurons, which project out cortically from the primary motor cortex to the brainstem before descending down the spinal cord [2]. Traumatic brain injury is often related to localized lesions that do not systematically involve primary lesions in the motor cortex. However, secondary brain lesions might arise in the motor cortex over time, due to diffuse axonal injury and degradation of white matter tracts, and lead to lesions that in turn would affect upper motor neuron integrity and increase spasticity [18]. In contrast, non-traumatic etiologies such as anoxia (which represents most patients in this subsample) affect the cortex more globally and might more frequently involve primary lesions in the motor cortex and upper motor neurons, leading more quickly to higher level of spasticity. For the lower limbs, the level of consciousness appears to be important, with patients in a VS developing more spasticity over time than patients in an MCS. The sustained immobilization and lack of voluntary movement that characterizes VS patients might explain why a higher level of spasticity was observed in the lower limbs [3]. In addition, the lower limbs may be more affected by gravity than the upper limbs, which affects bedridden patients and especially their ankles, as it increases the risk of developing equinovarus feet. In our sample, the cause of brain injury did not seem to significantly impact the development of spasticity in the lower limbs. 

These findings underline how the factors impacting spasticity over time differ in patients with prolonged DOCs, and how different spasticity should be addressed and treated in this challenging population. As mentioned in the introduction, there are only a few treatments that have shown efficacy in these patients. In our dataset, the main antispastic medicine was baclofen, and did not seem to impact changes in spasticity among patients. This is in agreement with the literature, where there is limited evidence of effective baclofen treatment of spasticity in individuals with a brain injury, despite its broad usage [9]. On the contrary, the use of behavioral interventions (such as soft splint and frequent physical therapy sessions) have shown some preliminary efficacy and might be helpful [10,11]. The efficacy of some pharmacological agents (e.g., monoaminergic drugs) that have an impact on motor cortex excitability and plasticity could be investigated [19]. The neuromodulation of M1 using non-invasive techniques such as transcranial direct current stimulation (tDCS) or transcranial magnetic stimulation (TMS) has also shown some promising results and should also be further assessed [20,21]. 

This study had several limitations. First, due to the retrospective nature of this study, a comprehensive list of factors that might influence spasticity could not be included (e.g., MRI results to identify exact brain lesions, or EMG recordings to better characterize profiles of spasticity). A more comprehensive set of variables should be considered in the future using a prospective design. Second, our sample size was small, and our results should be replicated in a bigger sample including various etiologies (such as stroke). Future data should be collected earlier during patients’ recovery to have a more comprehensive understanding of the evolution of spasticity over time. Third, the control of medication focused exclusively on the use of antispastic medication (e.g., baclofen). In the future, studies should address medications that have the potential to generate new spasticity or worsen current spasticity as a side effect. Finally, the patients’ level of consciousness was assessed with the WNSSP, which has been recommended with moderate reservation by the ACRM. The gold standard scale that is currently used to detect signs of consciousness among patients with prolonged DOCs is, nevertheless, the Coma Recovery Scale-Revised (CRS-R) which is recommended with minor reservation by the ACRM. The CRS-R should therefore be the scale of choice in future studies to characterize patients’ behavioral profile [15].

In summary, spasticity frequently develops in patients diagnosed with a DOC impacting the rehabilitation process and their quality of life. Our preliminary results suggest that underlying brain lesions and the level of consciousness might impact the development of spasticity over time differently. This study has important clinical implications, as treatment options are very limited and inefficacious at treating spasticity in DOC patients. Understanding the mechanisms of spasticity will most likely help clinicians in finding treatments that are more efficient to manage the disorder.

## Figures and Tables

**Figure 1 brainsci-12-00295-f001:**
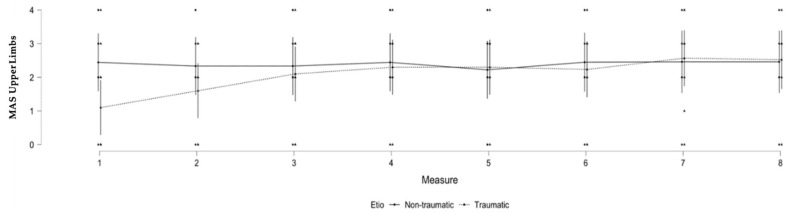
Significant changes in MAS scores according to etiology over time for the upper limbs. Legend: MAS = Modified Ashworth Scale, Etio = etiology.

**Figure 2 brainsci-12-00295-f002:**
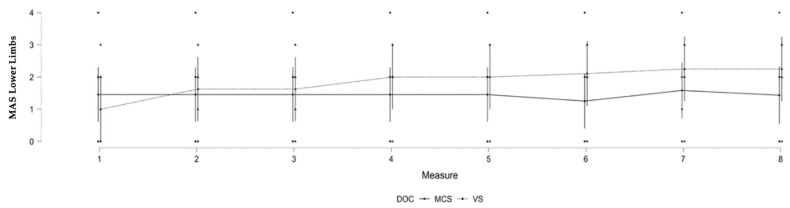
Significant changes in MAS scores according to level of consciousness over time for the lower limbs. Legend: MAS = Modified Ashworth Scale, DOC = disorder of consciousness (VS = vegetative state/unresponsive wakefulness syndrome, MCS = minimally conscious state).

**Table 1 brainsci-12-00295-t001:** Patients’ medical and demographic information.

Gender	Age	TSI	Etio	DOC	WNSSP	Medication	Number Measures
M	34	99	Traumatic	MCS	46	1	7
M	47	66	Traumatic	VS	0	0	8
M	23	404	Non-traumatic (anoxia)	MCS	15	0	8
M	25	1006	Traumatic	VS	13	1	8
F	31	389	Non-traumatic (anoxia)	VS	11	0.875	8
M	48	1022	Non-traumatic (anoxia)	MCS	14	0	6
M	67	37	Traumatic	MCS	21	0	8
M	57	93	Traumatic	VS	4	0.75	8
F	53	397	Non-traumatic (anoxia)	MCS	47	0	8
F	60	1118	Non-traumatic (anoxia)	MCS	21	0	5
M	56	286	Traumatic	VS	2	0	8
M	48	36	Non-traumatic (anoxia)	MCS	27	0	8
M	64	164	Traumatic	VS	4	0.8	5
M	47	87	Traumatic	MCS	21	0.857	7
M	65	91	Non-traumatic (anoxia)	VS	6	0.875	8
F	26	308	Non-traumatic (anoxia)	MCS	13	1	6
M	34	799	Non-traumatic (anoxia)	MCS	66	1	8
M	62	83	Traumatic	MCS	89	0	8
F	24	544	Traumatic	VS	7	0.125	8

Legend: Gender (M = male; F = female), Age (in years), TSI = time since injury (in days), Etio = etiology, DOC = disorder of consciousness (VS = vegetative state/unresponsive wakefulness syndrome, MCS = minimally conscious state), WNSSP = Western Neuro Sensory Stimulation Profile, Medication = ratio of the presence of antispastic drug on the number of assessments, Number Measures = number of MAS assessments.

**Table 2 brainsci-12-00295-t002:** Linear mixed models over the 8 assessments of MAS for the upper limbs.

MAS Evolution Upper Limbs			
Effect	*df*	F	*p*
Etio	7, 107.29	2.226	0.038 *
TSI	7, 107.38	0.695	0.676
Age	7, 107.13	0.159	0.992
Medication	7, 107.20	1.129	0.350
Gender	7, 107.54	0.359	0.924
DOC	7, 107.19	0.569	0.780

Legend: MAS = Modified Ashworth Scale, Etio = etiology, TSI = time since injury (in days), Age (in years), Medication = ratio of the presence of antispastic drug on the number of assessments, Gender (M = male; F = female), DOC = disorder of consciousness (VS = vegetative state/unresponsive wakefulness syndrome, MCS = minimally conscious state). * = significant results

**Table 3 brainsci-12-00295-t003:** Linear mixed models over the 8 assessments of MAS for the lower limbs.

MAS Evolution Lower Limbs			
Effect	*df*	F	*p*
Etio	7, 107.09	1.245	0.285
TSI	7, 107.12	0.531	0.809
Age	7, 107.07	0.533	0.808
Medication	7, 107.11	0.547	0.797
Gender	7, 107.15	0.402	0.899
DOC	7, 107.07	3.196	0.004 *

Legend: MAS = Modified Ashworth Scale, Etio = etiology, TSI = time since injury (in days), Age (in years), Medication = ratio of the presence of antispastic drug on the number of assessments, Gender (M = male; F = female), DOC = disorder of consciousness (VS = vegetative state/unresponsive wakefulness syndrome, MCS = minimally conscious state). * = significant results

## Data Availability

All data that support the findings of this study are available upon request from the corresponding author.

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
