# Peer review of "Changes of Spasticity across Time in Prolonged Disorders of Consciousness: A Retrospective Study"

_brainsci, 2022, doi:10.3390/brainsci12020295_

Round 1

Reviewer 1 Report

This is a highly readable paper that despite some limitations is clinically relevant and very informative. 

My main comment is that there are other factors that could influence spasticity that are not included in your study such as nutrition but also some medications (other than antispastic) so it would be useful to have one sentence to say that medication and feeding regime remained the same during the study (if that is correct). 

Author Response

This is a highly readable paper that despite some limitations is clinically relevant and very informative.  My main comment is that there are other factors that could influence spasticity that are not included in your study such as nutrition but also some medications (other than antispastic) so it would be useful to have one sentence to say that medication and feeding regime remained the same during the study (if that is correct). 

We thank reviewer 1 for their positive feedback and their comments. While feeding did not change due to the presence of g-tube in all patients (typical in chronic DOCs), medication has changed over 21months. Since other variables of interest were included in our statistical model, we had to make a choice regarding medication. We decided to focus on antispastic medication since this was the focus of our paper. However, we agree with the reviewer that the impact of other medications should further be investigated in the future and that this is a limitation of our study, which has been added as such in the Discussion. We also added that “All patients were provided nutrition by feeding tube” in the Methods (see Study population” subsection).

Reviewer 2 Report

Suggestions for Authors

Thank you very much for the opportunity to review the manuscript of B. Winters and colleagues! The authors provide valuable research regarding spasticity in patients with prolonged disorders of consciousness. This issue urgently needs to be investigated more closely due to the high prevalence and the very limited therapy options.

General comments

  • Each table and figure should have a legend explaining the used abbreviations and statistical tests / significance levels that are illustrated
  • Figure 1 and 2: can you highlight the significant differences? Please change the variable names “MAS_up” and “MAS_down”

Specific comments

  1. 3, l.99: “of the joints of the upper and lower limbs”
  2. 3., l.111 maybe you could add the term “Unresponsive Wakefulness Syndrome (UWS)” which is often used in DOC literature (I suppose you used VS because it is the diagnostic category used in the WNSSP?)

p.3, l.140: What were the reasons for the missing data?

p.4, Table 1: MAS is mentioned in the legend of the table but the corresponding data are missing in the table

p5.,l.172: “A significant interaction was found between DOC and only the lower limb spasticity” -> is “only” correct (since the subheading is “lower limb spasticity”)

p.5,l.175: “at the first time-point” -> is this correct? You mentioned before that VS patients developed more spasticity in the 2nd year of observation

p.5, Table 3: DOC should be highlighted instead of Etio (=significant effect)

p.6, l.196: did not seem to be related?

p.6., l.208: through

p.6, l. 215f.: These findings underlined … and underlined… (I would either delete the second use of the word “underlined” or use another word instead)

Author Response

Thank you very much for the opportunity to review the manuscript of B. Winters and colleagues! The authors provide valuable research regarding spasticity in patients with prolonged disorders of consciousness. This issue urgently needs to be investigated more closely due to the high prevalence and the very limited therapy options.

General comments: 1) Each table and figure should have a legend explaining the used abbreviations and statistical tests / significance levels that are illustrated; 2) Figure 1 and 2: can you highlight the significant differences? Please change the variable names “MAS_up” and “MAS_down”

Thank you for pointing this out. We added legends for each table and figure. We highlighted significant differences in bold in the tables (except for graphs since LMM does not compare each time-point at a time but rather look at the curve of change through time). Finally, we changed the variable names in the figures to MAS Upper Limbs and MAS Lower Limbs.

Specific comments:

  1. , l.111 maybe you could add the term “Unresponsive Wakefulness Syndrome (UWS)” which is often used in DOC literature (I suppose you used VS because it is the diagnostic category used in the WNSSP?)

The term UWS has been mentioned in the text (as well as in our list of keywords). We also added the reference: Laureys S, Celesia GG, Cohadon F, et al. Unresponsive wakefulness syndrome: A new name for the vegetative state or apallic syndrome. BMC Medicine 2010; 8:68.

  1. 3, l.140: What were the reasons for the missing data?

Attrition was due to the transfer to another long-term care facility or discharge home. We included this information in the Results (see “Participants” subsection).

  1. 6, l.196: did not seem to be related?

We clarified the wording. You can now read: “As per our statical analyses, this difference were not related to other factors such as age, time since injury, the presence of spastic medication or the level of consciousness at time point 1”.

  1. 3, l.99: “of the joints of the upper and lower limbs” AND 4, Table 1: MAS is mentioned in the legend of the table but the corresponding data are missing in the table AND p5.,l.172: “A significant interaction was found between DOC and only the lower limb spasticity” -> is “only” correct (since the subheading is “lower limb spasticity”) AND p.5, Table 3: DOC should be highlighted instead of Etio (=significant effect) AND p.6., l.208: through AND p.6, l. 215f.: These findings underlined … and underlined… (I would either delete the second use of the word “underlined” or use another word instead) AND p.5,l.175: “at the first time-point” -> is this correct? You mentioned before that VS patients developed more spasticity in the 2nd year of observation

These were typos that were corrected. We thank the reviewer for pointing these out.